# Membrane Lipid Derivatives: Roles of Arachidonic Acid and Its Metabolites in Pancreatic Physiology and Pathophysiology

**DOI:** 10.3390/molecules28114316

**Published:** 2023-05-24

**Authors:** Cándido Ortiz-Placín, Alba Castillejo-Rufo, Matías Estarás, Antonio González

**Affiliations:** Instituto de Biomarcadores de Patologías Moleculares, Departamento de Fisiología, Universidad de Extremadura, 10003 Cáceres, Spain; coplacin@unex.es (C.O.-P.); alcasru@unex.es (A.C.-R.); meh@unex.es (M.E.)

**Keywords:** arachidonic acid, cancer, inflammation, fibrosis, pancreas, stroma

## Abstract

One of the most important constituents of the cell membrane is arachidonic acid. Lipids forming part of the cellular membrane can be metabolized in a variety of cellular types of the body by a family of enzymes termed phospholipases: phospholipase A2, phospholipase C and phospholipase D. Phospholipase A2 is considered the most important enzyme type for the release of arachidonic acid. The latter is subsequently subjected to metabolization via different enzymes. Three enzymatic pathways, involving the enzymes cyclooxygenase, lipoxygenase and cytochrome P450, transform the lipid derivative into several bioactive compounds. Arachidonic acid itself plays a role as an intracellular signaling molecule. Additionally, its derivatives play critical roles in cell physiology and, moreover, are involved in the development of disease. Its metabolites comprise, predominantly, prostaglandins, thromboxanes, leukotrienes and hydroxyeicosatetraenoic acids. Their involvement in cellular responses leading to inflammation and/or cancer development is subject to intense study. This manuscript reviews the findings on the involvement of the membrane lipid derivative arachidonic acid and its metabolites in the development of pancreatitis, diabetes and/or pancreatic cancer.

## 1. Introduction

This work is an invited review that covers interesting findings on the involvement of the membrane lipid derivative arachidonic acid and its metabolites in the development of pancreatic diseases such as pancreatitis, diabetes and/or pancreatic cancer. Released from the constituent lipids of the cellular membrane, arachidonic acid is metabolized by major enzymes, which yields a subfamily of derivatives with important functions in cell physiology and pathophysiology, either in an autocrine, paracrine or endocrine manner. In this regard, the so-called tumor microenvironment (TME) has been signaled as an important cause of pancreatic disease. Specifically, the TME plays a critical role in cancer proliferation, invasion, metastasis and resistance to radiotherapy and chemotherapy. Because the relationship between inflammation and cancer is close, the role of the stroma and the putative cell-to-cell intercommunication in the pathophysiology of the pancreatic gland has been extensively analyzed. This research tries to summarize the advances made in the involvement of membrane lipid derivatives in pancreatic physiology and focuses its attention on the usefulness of arachidonic acid (AA) metabolism for the diagnosis, prevention and/or treatment of pancreatic disorders.

## 2. Arachidonic Acid Metabolism in the Pancreas

Lipid-derived messengers are well-known mediators of intracellular signaling. One of the most important constituents of biological membranes is AA, a polyunsaturated 20 carbon fatty acid (PUFA). Due to the presence of four double bonds in its molecule, AA can interact with oxygen molecules in order to generate several bioactive compounds such as eicosanoids and isoprostanes [1]. Normally, membrane-embedded lipids can be metabolized in a variety of cellular types of the body by a family of enzymes termed phospholipases. These enzymes are mostly responsible for the release of AA from glycerophospholipids located within the cellular membrane. Three types of phospholipases have been described: phospholipase A2 (PLA_2_), phospholipase C (PLC) and phospholipase D (PLD). PLA_2_ is considered the most important enzyme type for the metabolization of AA. It recognizes the sn-2 acyl bond within the membrane phospholipids and releases AA and lysophospholipids in a single step [2,3,4]. The mentioned enzyme comprises six subtypes, which are known as cytosolic PLA_2_, calcium-independent PLA_2_, secreted PLA_2_, lysosomal PLA_2_, platelet-activating factor acetyl-hydrolases and adipose-specific PLA_2_ [5,6]. AA exerts a pivotal role as a messenger in the cell physiology [7]. In addition to its role as an intracellular signal, AA also acts as a precursor for eicosanoid synthesis. The latter, in turn, also plays biological roles in the body, including the pancreas [8,9]. In this context, AA can be metabolized via three possible enzymatic pathways, through the enzymes cyclooxygenase (COX), lipoxygenase (LOX) and cytochrome P450 (CYP450) [10].

COX is also known as a prostaglandin endoperoxide synthase and is a type of oxidoreductase enzyme. Two isoforms of COX exist, termed COX-1 and COX-2. COX-1 is constitutively expressed in nearly all tissue and carries out normal (physiological) functions. COX-2 is inducible. The inducing stimuli include pro-inflammatory cytokines and growth factors. Consequently, COX-2 has been signaled to play a key role in inflammation and the control of cell growth [11]. COX-2 is highly expressed in a variety of cancers, where it exerts pleiotropic and multifaceted functions, which are considered responsible for, or critical contributors to, the genesis and/or promotion of carcinogenesis and cancer cell resistance [12]. COX generate prostanoids (20-carbon fatty acids), including prostaglandins (PGs) and thromboxanes (TXs). AA is firstly cyclized and oxygenated by COX. This step forms a cyclic endoperoxide derivative, prostaglandin G2 (PGG2). The hydroperoxyl group of PGG2 is also rapidly reduced by COX to produce hydroxyl-prostaglandin H2 (PGH2) [13]. PGH2 is later metabolized by various downstream enzymes, such as prostaglandin synthases and isomerases, leading to the generation of other specific PGs in the pancreas, such as prostaglandin E2 (PGE2—the most abundant PG in humans), prostaglandin D2 (PGD2), prostaglandin F1 alpha (PGF1α) and prostacyclin or PGI2. COX isoform derivatives have a preference for downstream PG synthases/isomerases. However, this coupling is not exclusive. For example, COX-1-derived PGH2 pairs mainly with PGF synthase, thromboxane synthase and cytosolic PGE synthase. Meanwhile, COX-2-derived PGH2 feeds mostly with PGI synthase and microsomal PGE synthase [14,15,16]. PGs exert their effects by activating membrane-linked G-protein receptors and have several different effects, depending on the specific PG and the cell type [17]. Another COX isoform has been found, COX-3, which is produced by the splicing of COX-1. However, its expression in the pancreas has not been properly studied yet. The only information that we could find refers to the study by Persaud et al. [18], who signal that this form of COX is not expressed in pancreatic β cells. Likewise, PGH2 can be metabolized to TXA2 by the TXA2 synthase. The latter is a highly unstable compound; its half-life is approximately 30 s. Thus, it is spontaneously hydrolyzed to thromboxane B2 (TXB2) [15,19]. TXB2 or its derivates, 2,3-dinor-TXB2 and 11-dehydro-TXB2, can be used as parameters for TXA2 production, whose increase is associated with pancreatic diseases [20,21,22].

LOX comprises a group of dioxygenases that catalyze the peroxidation of PUFAs, mainly linoleic acid (LA) or AA, in the presence of molecular oxygen [23]. The LOX family is classified according to the carbon atom that is oxygenated. Four types of LOX have described: 5-lipoxygenase (5-LOX), 8-lipoxygenase (8-LOX), 12-lipoxygenase (12-LOX) and 15-lipoxygenase type 1 and 2 (15-LOX-1 and 15-LOX-2). However, the main enzymes expressed in humans are 5-LOX (called leukocyte 5-LOX), 12-LOX (also known as platelet 12-LOX and leukocyte 12-LOX, respectively) and 15-LOX [3,24]. Generally, these enzymes synthesize active metabolites such as leukotrienes (LTs), lipoxins (LXs), hepoxillins (HOs) and hydroxy-eicosatetraenoic acids (HETEs) [25], which could act in an autocrine, paracrine and/or endocrine manner [26]. LOX enzymes generate the forms 5-, 8-, 12- or 15- hydroperoxyl-eicosatetraenoic acids (HpETEs), which are reduced by glutathione peroxidase (GPx) to the hydroxy forms (5-, 8-, 12-, 15-HETE), respectively [3,25]. However, the major sites on which AA is oxygenated are represented by the 5-, 12- and 15-positions. Therefore, in this review, we will focus on 5-LOX, 12-LOX and 15-LOX, which have been indicated to play important roles in the development and progression of human cancers, including pancreatic cancer, while 8-LOX has not been properly studied in the pancreas yet [10,24,27].

Firstly, 5-LOX generates 5-HpEPES from AA, which is a precursor for the synthesis of leukotrienes A4 (LTA4), 5-HEPEs and lipoxins (LXA4 and LXB4) [14]. LTA4 is an unstable metabolite that might be converted into LTB4 by LTA4 hydrolase. Then, LTB4 is conjugated with glutathione by glutathione S-transferase (GST) in order to produce LTC4, LTD4 and LTE4 [24]. With regard to the 12-LOX pathway, the metabolite generated from AA is termed 12-HpETES, which is further reduced to the more stable form 12-HETE. In addition, 12-HETE could lead to the generation of HO forms, such as 8-hydroxy-11,12-epoxy-eicosatetraenoic acid (HxA3) and 10-hydroxy-11,12-epoxy-eicosatrienoic acid (HxB3). Finally, the 15-LOX enzyme is subdivided into two subtypes, 15-LOX-1 and 15-LOX-2, which may have pro- or antitumorigenic activity depending on the isoforms. Moreover, 15-LOX-1 metabolizes AA to 3-hydroxy-octadecadienoic acid (13-HODE) and 15-HETE, while 15-LOX-2 mainly produces 15-HETE. Then, 15-HETE, which is a substrate for 5-LOX, is rapidly converted to LXA4 and LXB4 [28].

CYP450 consists of a group of membrane-bound hemoproteins with enzymatic activity that detoxify xenobiotics and exert key roles in cellular metabolism and homeostasis. CYP450 can be transcriptionally activated by different xenobiotics and endogenous substrates and its expression can be modulated by hormones and growth factors [29]. The CYP family is composed of numerous subclasses, giving rise to more than 6000 different enzymes. However, the CYP enzymes that catalyze AA are those with omega-hydroxylases and epoxidase activity. The former enzymes provide HETEs from AA and the latter produce epoxides and epoxyeicosatrienoic acids (EETs). The best known HETEs are 6-, 12-, 17-, 18-, 19- and 20-HETE. Some HETEs synthesized by CYP450 can be transformed subsequently by LOX [30]. CYP450 epoxygenases generate four EETs: 5,6-EET; 8,9-EET; 11,12-EET; and 14,15-EET. Stereoisomers exist for each of the four EET regioisomers, each of which may exert different effects [31]. CYP450 2J2 (CYP2J2) is the primary extrahepatic enzyme that processes AA to produce EETs in human [32]. These EETs have functions of their own, such as promoting angiogenesis and cell migration and cancer. In addition, EETs are rapidly metabolized by epoxide hydrolases to form dihydroxyeicosatrienoic acids (DHETs), which also exhibit diverse cellular functions [33,34,35]. The rapid transformation of EETs into DHETs has made it necessary to inhibit or delete epoxide hydroxylase to analyze the effects of EETs [36].

In summary, all these enzymes convert AA into different metabolites that will be involved in cellular fates. Among them, PGs, TXs, LTs and HETEs are the major metabolites generated from AA that are responsible for cellular responses [3]. Figure 1 provides a schematic representation of the bioactive eicosanoids derived from the AA pathway.

Finally, the activation of cell membrane receptors for pancreatic secretagogues, such as cholecystokinin, is linked to PLA_2_ and to AA metabolism and, hence, modulates pancreatic function [18,37]. Furthermore, the overstimulation of pancreatic cells with secretagogues has been related to pancreatic disease [38,39]. Thus, either unbalanced secretagogue stimulation or the arrival of noxious agents may alter signaling in pancreatic cells, which has been related to the onset of pancreatic diseases such as inflammation and cancer [40,41,42,43,44,45].

In line with the above-reported information, AA and its metabolites, as well as associated genes, could be used for prognostication and therapeutic aspects of pancreatic diseases. For example, an impaired serum fatty acid composition has been associated with primary insulin autoimmunity, on the basis that higher palmitoleic acid, cis-vaccenic, arachidonic, docosapentaenoic and docosahexaenoic acids decrease the risk, whereas higher α-linoleic acid and arachidonic:docosahexaenoic and n-6:n-3 acid ratios increase the risk [46]. Similarly, the lipidomic profiling of serum and pancreatic fluid can be used for the detection of chronic pancreatitis, a disease in which the serum and/or pancreatic fluid levels of oxidized fatty acids are elevated [47]. With regard to prognosis and therapy, signaling via 12-lipoxygenase-Gpr31 has been proposed to be required for pancreatic organogenesis in the zebrafish. This was based on the observation that 12-LOX-generated metabolites of AA increased sharply during organogenesis stages and that either the depletion or inhibition of 12-LOX impaired exocrine pancreas growth and the generation of insulin-producing β cells [48]. Additionally, the feedback modulation of glucose-induced insulin secretion by AA metabolites has been reported. In this context, the stimulation of insulin release by glucose may trigger a negative feedback loop via the local release of an inhibitor of β cell function. One or more metabolites of AA could be involved. Hence, the development of selective inhibitors of AA metabolism might be used in the therapy of impaired β cell function [49]. Moreover, the endocannabinoid system, which is reported as a lipid signaling system, comprises endogenous cannabis-like ligands that are derived from AA—for example, 2-arachidonoylglycerol. These signaling molecules bind to G-protein-coupled receptors, termed CB1 and CB2. The receptor for CB1 is widely distributed in the body, including the pancreas, where it could be coupled to the functional regulation of the gland and could represent a therapeutical approach [50]. Last but not least, suppression of the 5-LOX gene could be involved in the activation of apoptosis in pancreatic tumor cell lines in response to triptolide. These results suggest that the inhibition of AA metabolism by the 5-LOX pathway is associated with the antiproliferation activity of the mentioned drug, exhibiting clinical therapeutic value for patients with pancreatic cancer [51].

## 3. Arachidonic Acid and Calcium Signaling in the Pancreas

Calcium (Ca^2+^) is a universal intracellular messenger that, in addition, exhibits enormous versatility. As such, Ca^2+^ serves as a powerful tool to control a wide array of processes—for example, proliferation, growth and development, learning and memory, muscle contraction, fertilization and/or secretion (enzymes and hormones). Nevertheless, impairment of its control may lead to undesirable effects, which might result in cell transformation and/or cell death [52].

As in other tissues and organs, Ca^2+^ is a critical intracellular messenger in pancreatic cell physiology [53]. Moreover, impairment of Ca^2+^ homeostasis is of critical relevance for pancreatic diseases, such as inflammation and cancer [54,55]. Attention has been paid to how genetic alterations can conduct Ca^2+^ signaling pathways, in the sense that mutations or the impaired expression of critical proteins that are involved in the control of the intracellular Ca^2+^ concentration could lead to the deregulation of Ca^2+^-dependent effectors and, as a consequence, to the impairment of the mechanisms that control the cell’s fate. Of major relevance is the way in which the impairment of Ca^2+^ homeostasis might influence a cell´s behavior to promote enhanced proliferation, survival and invasion, which are all pathophysiological hallmarks of cancer [56,57].

Interestingly, the involvement of AA metabolism and Ca^2+^ signaling in the pancreas has been reported. The release of Ca^2+^ from its intracellular stores depends on the activation of different intracellular pathways by the gut hormone cholecystokinin, which may bind to low- and/or high-affinity receptors. The latter would be linked to the activation of PLA_2_ cascades. The products of the mentioned enzyme, AA and/or its metabolites, might modulate Ca^2+^ signals and, hence, the pancreatic physiology. Specifically, AA modulated the propagation of Ca^2+^ waves in pancreatic acinar cells, reducing the speed of propagation of Ca^2+^ signals [37,58,59]. Moreover, endogenously generated AA inhibited polyphosphoinositide synthesis and blocked agonist-induced inositol trisphosphate synthesis and Ca^2+^ mobilization in healthy cells [8].

On the contrary, AA might behave differentially in tumoral/transformed cells. In the study by Wu et al., the authors signaled that the AA pathway was involved in the increase in the intracellular Ca^2+^ concentration in pancreatic cancer cells. In the research, it was suggested that the metabolites of arachidonic acid were not involved in AA-mediated Ca^2+^ release. Rather, AA itself was responsible for their observation [60]. In rat pancreatic β cells, AA released Ca^2+^ from intracellular stores, an effect that was not observed in response to the analogue eicosatetraynoic acid. This further confirmed that AA metabolism was not required for Ca^2+^ mobilization [9]. Similarly, the analogue of AA arachidonyltrifluoromethyl ketone increased cytosolic Ca^2+^ in HIT insulinoma cells, which contributed to insulin secretion [61]. Thus, the effects of AA on cell physiology and its consequences might depend on the cellular type and on its status.

## 4. Role of Arachidonic Acid in Pancreatitis and Diabetes

As mentioned above, not only AA itself but its metabolization into different derivatives might be involved in major inflammatory responses in different tissues and organs, including the pancreas. Pancreatitis is a multifactorial disease that may be caused by the activation of different inflammatory mediators. Gallstones, smoking and alcohol consumption are well-established risk factors that may induce pancreatitis [62,63], but there are other factors contributing to the development of inflammation within the gland, including certain drugs such as mesalazine, azathioprine and simvastatine [64]. Additionally, post-endoscopic retrograde cholangiopancreatography has also been considered a potential risk factor that might induce pancreatitis as a common complication [65], in a similar way to pancreaticoduodenectomy [66]. Alteration of the immune response induced by gene mutations and/or environmental factors has also been considered a determinant of pancreatic damage [67]. All these factors might be related to the inappropriate intrapancreatic activation and release of pancreatic hydrolases. This could represent a pathogenetic mechanism of autodigestion of the gland that might lead to the onset of pancreatic inflammation [68]. The underlying cause responsible for the impairment of enzyme secretion could involve intracellular Ca^2+^ accumulation and concomitant oxidative stress [69,70,71]. The relationship between AA and Ca^2+^ signaling has been studied (see above).

In general, a major consensus points towards AA derivatives as being responsible for inflammation. COX, LOX and/or CYP450 produce metabolites that, to a variable extent, are involved in cell damage [72,73]. In fact, the levels of leukotriene B4 (LTB4), 15 hydroxyeicosatetraenoic acid (15-HETE), 6-keto prostaglandin F1 alpha (6-keto PGF1α), thromboxane B2 (TXB2) and prostaglandin E2 (PGE2) were increased in pancreatic tissue upon the induction of acute pancreatitis [20].

As previously reported, there are two genes encoding two COX isoenzymes. COX-1 is expressed constitutively in the cell and appears to regulate many normal physiologic functions. Conversely, COX-2 is inducible. Various growth factors, endotoxins, mitogens and tumor agents lead to an increase in its expression [74]. Activation of COX-2 has been signaled to mediate the inflammatory response. As such, COX-2 might be involved in the progression of inflammatory disease and in the development of chronic pancreatitis and/or diabetes [75]. COX-1 and COX-2 catalyze the formation of prostaglandins (PGs), thromboxanes (TXs) and levuloglandins [12]. The work carried out by Huang et al. showed that the expression of COX-2 in transgenic mice induced progressive changes in the pancreas, which included pancreas enlargement, inflammation, collagen deposition and acinar-to-ductal metaplasia [76]. Indeed, the success of anti-inflammatory drugs has been linked to their ability to inhibit COX-2 at the sites of damage and to a decrease in inflammation [77,78].

The generation of PGE2 through the COX pathway was considered a significant factor involved in β cell dysfunction and destruction and contributed to the pathogenesis of diabetes [3]. PGE2 is upregulated in diabetes and induces cell damage. Blockade of its receptor EP3 promoted β cell proliferation and survival via activation of the transcription factor nuclear factor E2-related factor 2 (Nrf2) [24]. AA metabolization led to the generation of PGE2 and TXB2, which activated the pro-inflammatory pathways nuclear factor kappa-light-chain-enhancer of activated B cells (NFκB) and Janus kinase (JAK)/signal transducer and activator of transcription 3 (STAT3) [79].

The induction of pancreatitis was related to increases in the levels of PGE2, PGD2 and TBX2, among others, in the gland [80]. High levels of PGE2 were detected in Sprague-Dawley rats upon the induction of acute necrotizing pancreatitis. Lipid peroxidation, together with edema, inflammation, bleeding and necrosis, was observed. The histopathologic severity was decreased by n-3 fatty acids, which was explained by the inhibition of PGE2 [81]. PGE2 modulated the activation of tumor necrosis factor alpha (TNF-α) in rat pancreatic lobules, an effect probably mediated by the activation of protein kinase A. This was interpreted as a putative mechanism that might explain the COX-2-dependent propagation of pancreatic inflammation [82]. Similarly, increased levels of PGE2 and TNF-α were detected in a cell line and animal models of severe acute pancreatitis. Meanwhile, 2-acetoxy-5-(2-4-(trifluoromethyl)-phenethylamino)-benzoic acid exhibited antioxidative and anti-inflammatory activity and diminished serum amylase and lipase levels and pancreatic wet weights, thereby inducing significant tissue-protective effects [83]. Rofecoxib, a selective COX-2 inhibitor, diminished PGE2 levels and collagen and transforming growth factor β (TGFβ) synthesis in an animal model of chronic pancreatitis. All the effects reported were related to the diminished infiltration of macrophages [84].

Apart from prostaglandins, leukotrienes also may be involved in the onset of pancreatic damage. LOX is a family of iron-containing dioxygenases that catalyze the formation of bioactive HETE metabolites from polyunsaturated fatty acids such as linoleic acid and AA [85]. These enzymes comprise six isoforms, which are expressed in different types of cells and tissues, such as immune, epithelial and tumor cells. LOX displays a wide range of functions, which include inflammation and tumorigenesis. The major end products of their activity are termed leukotrienes [86]. It should be pointed out that lipoxins have an important role in cancer cells. Lipoxins have anti-inflammatory effects, which decrease the chronic inflammation in the damaged tissue. This action takes place when these molecules bind to G-protein-coupled lipoxin A4 receptor (ALX)/formyl peptide receptor (FPR2). On the one hand, lipoxins may interact with many cells of the immune system, such as neutrophils, macrophages and T and B cells. On the other hand, these molecules also might regulate the levels of several transcription factors, such as NFκB factor [87]. Specifically, it has been shown that LXA4 is able to reduce cell proliferation, to inhibit cell invasion and to suppress tumor growth, therefore exhibiting anti-inflammatory properties in cancer. For this reason, it is important to evaluate its putative use in the treatment of cancer [25].

Caerulein, a pancreatic secretagogue, or intraductal bile acids, are common tools used to induce pancreatitis. Both of them increased the production of LTB4 in mice pancreatic acinar cells. In the study, a marked increase in the level of 5-LOX was observed [88]. The administration of LTB4 induced pancreatic damage, evidenced by pancreatic edema, neutrophil infiltration and necrosis [89]. LTB4 induced polymorphonuclear leucocyte accumulation in response to the generation of oxygen free radicals in an experimental rat model of pancreatic inflammation. The LTB4 inhibitor MK-886 exhibited protective effects [90]. The 5-LOX inhibitor zileuton repressed blood biomarkers of neutrophil activation and attenuated pancreatic tissue damage in a rat model [91]. Zafirlukast, a leukotriene receptor antagonist, improved histopathological parameters in the pancreas of rats subjected to acute pancreatitis [92].

Moreover, 12-lipoxygenase (12-LOX) has also been involved in the development of diabetes [26]. Increases in the levels of 12- and 15-HETE followed increases in the expression of 12/15-LOX and were accompanied by islet dysfunction and insulin resistance [93]. Leukotrienes inhibited glucose-induced insulin release, thereby altering endocrine pancreas functioning [94]. Zafirlukast also proved to be a potential candidate for a therapeutic intervention in diabetes, since it enhanced insulin secretion and prompted the activation of Ca^2+^/calmodulin-dependent protein kinase II and extracellular signal-regulated kinase signaling. Zafirlukast treatment further resulted in a significant drop in glucose levels [95].

CYP450 is a group of enzymes that are differentiated by a number for the isoform or individual enzyme (e.g., CYP1A1, CYP2D6) [96]. The major site of CYP expression is the liver [97]. These enzymes convert AA to four EETs that exhibit various biological effects, especially in the cardiovascular system [32]. There is some evidence for the involvement of CYP in pancreatic β cell dysfunction. CYP1A1 and CYP1B1 have been signaled to be involved in glucose homeostasis, insulin resistance and diabetes development [98]. EETs play an important role in insulin and glucagon secretion. Moreover, 5,6-EET was found to increase insulin release, while 8,9-, 11,12- or 14,15-EET stimulated glucagon production [99]. Loss of epoxide hydroxylase, an enzyme that degrades EETs, significantly reduced hyperglycemia in streptozotocin-treated mice and increased glucose-dependent insulin secretion and reduced apoptosis in pancreatic β cells [36]. An association between the CYP27B1 and CYP24A1 gene polymorphisms and type 1 diabetes has been reported [100]. However, to our knowledge, studies on the involvement of CYP in pancreatitis are currently lacking.

In general, it is accepted that the inhibition of the AA pathway and/or blockade of its metabolites will undoubtedly protect the pancreas against inflammation. In this context, knockout of the co-chaperone protein St13 exacerbated fatty replacement and fibrosis in a model of chronic pancreatitis. Therefore, it was suggested that St13 was functionally activated in acinar cells and exerted protection against inflammation, via regulation of the AA pathway [101].

Although the majority of the studies reviewed point toward the deleterious actions of AA metabolism, protective actions of AA have been reported in the pancreas. AA restored the antioxidant status to a normal range in an experimental animal model of diabetes mellitus. The study suggested that AA exerted protective actions in pancreatic β cells against alloxan-induced diabetes by attenuating oxidative stress [102]. The AA present in ARASCO oil was related to protective actions in the pancreas. Indeed, ARASCO oil, which was identified as a source of AA, diminished hyperglycemia, restored insulin sensitivity, suppressed inflammation and reversed the altered antioxidant status in streptozotocin-induced diabetes mellitus [103]. In a similar way, a decrease in PGE2 was correlated with an increase in tissue damage in alcohol-fed rats. Conversely, there was an inverse correlation between PGE2 levels and fibrogenesis. Because of these observations, the authors of this research suggested that endogenous PGE2 plays a protective role in alcohol-induced injury in the pancreas [104]. All this suggests the possibility, mentioned earlier in this manuscript, that the effects of AA on cell physiology and its consequences may vary from cell to cell and also may depend on the cellular state.

## 5. Arachidonic Acid’s Involvement in Pancreatic Cancer Development

The relationship between inflammation and cancer is widely accepted [105], including in pancreatic cancer [106]. In fact, both COX-2 and 5-lipoxygenase (5-LOX) are upregulated in different types of cancer, including pancreatic cancer. It is generally accepted that the metabolization of AA by these two enzymes leads to the formation of eicosanoids that directly contribute to pancreatic cancer cell proliferation, whereas the inhibition of the mentioned enzymes abrogates cancer cell proliferation [107].

The COX-2 enzyme appears to be overexpressed in a number of cancers and, as such, its products might play critical roles in carcinogenesis [108]. In fact, COX-2 activation has been related to an increase in cell proliferation and survival and the inhibition of the pro-apoptotic pathway, thereby resulting in tumor angiogenesis, invasion and metastasis [109]. Thus, the inhibition of COX-2 has received increasing attention as a useful tool for the prevention and treatment of cancer [110,111,112]. Drugs that blocked COX enzymes inhibited pancreatic cancer growth both in vitro and in vivo and induced cell death through the activation of apoptosis [113]. Omura et al. [114] suggested that COX-2-derived PG are used by cancer cells to proliferate and that the inhibition of its import from the extracellular medium via the blockade of multidrug-resistance-associated proteins abrogated pancreatic cancer growth. Likewise, the expression of PG biosynthetic pathway enzymes in mucinous pancreatic cysts has been detected. Additionally, the levels of COX-2 and cPLA_2_ were increased in the epithelia of mucinous pancreatic cysts [115]. Concomitant activation of COX-2 and K-Ras(G12D) accelerated the progression of pancreatic intraepithelial lesions, which involved components of the neurogenic locus notch homolog protein 1 (Notch1) [116]. The expression of COX-2 was increased in a number of resection specimens of pancreatic ductal adenocarcinoma (PDCA). Synergistic increases were also detected in the expression of the tumor protein p53 [117]. The latter consists of a gene that codes a protein (p53) that plays a key role in controlling cell division and cell death and, hence, is pivotal to cancer cell growth and the development of the disease [118]. Therefore, the inhibition of COX-2 and/or PLA_2_ might help to prevent the progression of tumor cell growth and cancer development [115]. A decrease in both cell proliferation and PG levels was related to the improved turnover of preneoplastic lesions in the pancreas [119]. The COX inhibitor ibuprofen exerted modulating effects in pancreatic cancers experimentally induced in hamsters [4].

With respect to LOX enzymes, evidence also exists that signals their involvement in pancreatic cancer development. Expression of both 5-LOX and 12-LOX has been detected in pancreatic cancer tissue [113] and cell lines, including PANC-1, AsPC-1 and MiaPaCa2 cells. The expression of the receptor of the downstream 5-LOX metabolite, leukotriene B4, was also increased [120]. LOX metabolites stimulated the growth of the tumor cells, whereas their inhibition markedly inhibited pancreatic cancer cell proliferation [113]. Similarly, growth inhibition and apoptosis in human pancreatic cell lines were noticed upon the downregulation of 5-LOX expression, as well as in the production of its downstream product LTB4 [51]. Meanwhile, 15-LOX-1 expression and activity were suggested to exert antitumorigenic effects in pancreatic cancer. Interestingly, 15-LOX-1 was found to be strongly present in normal ductal cells, tubular complexes and centro-acinar cells, whereas no staining was noted in tumor cells. These observations suggested that 15-LOX-1 expression might be lost during pancreatic cancer development [121]. The lichen (symbiotic partnership of a fungus and an alga) metabolites protolichesterinic acid, lobaric acid and baeomycesic acid exhibited antiproliferative effects against twelve different human cancer cell lines, including the pancreatic cancer cell lines Capan-1, Capan-2 and PANC-1. All three compounds had in vitro 5-LOX inhibitory activity, whereas protolichesterinic and lobaric acid inhibited 12-LOX [122]. Zyflo, a selective inhibitor of 5-LOX, diminished the incidence and the sizes of carcinomas in a model of pancreatic cancer induced in Syrian hamsters. Additionally, the activity of several antioxidant enzymes was increased and the concentration of products of lipid peroxidation was decreased. Thus, it was suggested that the inhibition of 5-LOX might be useful to decrease tumor growth in advanced pancreatic cancer [123]. Moreover, Zyflo also diminished liver metastasis in pancreatic cancer [124]. Another compound, termed triptolide, induced apoptosis that was related to the inhibition of the 5-LOX pathway in SW1990 pancreatic cancer cells in vitro. Conversely, overexpression of 5-LOX or the exogenous administration of LTB4 made cells more resistant [51]. The work carried out by Tersey et al. showed that 12-LOX promoted inflammation and increased the production of reactive oxygen species, through the p38 mitogen-activated protein kinase (p38 MAPK) and NFκB pathway [125]. Therefore, 5-HETE and 12-HETE metabolites might promote cancer growth via activation of the p44/42 and PI3/Akt mitogen-activated protein kinase pathways [27].

CYP450s are considered another group of enzymes related to AA metabolism that play a key role in carcinogenesis. They represent a superfamily of enzymes that catalyze the oxidation of lipids, steroids and drugs [126]. Several CYP enzymes metabolically activate procarcinogens [127]. The expression of the form CYP2J2 is increased in various human tumor cells and its metabolites are suggested to be involved in the development of human cancers [32]. CYP2J2 is overexpressed in PDAC, and its metabolites downstream of AA, specifically 8,9-EET, could inhibit ferroptosis in the pancreatic tumor line PANC-1 [34]. CYP450-2E1 (CYP2E1) was induced by ethanol consumption and was related to consequent toxicity, including carcinogenesis in the gastrointestinal tract [128]. The expression of certain CYP4 isoforms has been detected at significantly high levels in PDAC tissue, suggesting a role in the development of the disease [126]. The role of this enzyme system in the pathogenesis of chronic inflammatory and malignant pancreatic diseases has been confirmed by other studies. Compared to the normal pancreas, the expression of CYPs was increased in a number of pancreatic cancer samples [129]. CYP2A6, a metabolic enzyme that activates several procarcinogens, which include dietary and tobacco-specific nitrosamines, has been linked to pancreatic cancer. Consistent with this, high levels of CYP2A6 activity were detected in patients suffering from pancreatic cancer [130].

## 6. Arachidonic Acid and Stroma: Interplay in the Tissue Microenvironment

The stroma comprises the cells, components and structures that support and give form to organs, glands and/or tissues in the body. The stroma is mainly composed of connective tissue, blood vessels, lymphatic vessels and nerves. It provides the conditions for the normal functioning of the cells present in a certain tissue or organ [131]. Replacement of the exocrine parenchyma (i.e., the tissue formed by the cells that carry out an essential function) by fibrous tissue is a major characteristic of chronic pancreatitis and pancreatic cancer [76].

The tumor microenvironment (TME) is defined as the environment and/or conditions that surround a population of cells that form a tumor [132]. This environment includes blood vessels, immune cells, fibroblasts, stellate cells, signaling molecules, respiratory gases and the extracellular matrix, all of which participate as constituents of the stroma [133]. In cancer, the stroma plays an important role in contributing to the development of malignant cells [131]. The cells and their surroundings establish a close relationship, i.e., a bidirectional interaction is constantly ongoing “indoors”. This creates favorable conditions that will confer upon tumor cells the capability for active proliferation and growth, invasion and metastasis, which are all hallmarks of malignancy [134].

As previously reported, inflammation within the tumor microenvironment is a hallmark of cancer and is accepted as a major characteristic of carcinogens. The participation of AA´s metabolites has been investigated. Its products, and the metabolism of related fatty acids, which include prostaglandins, leukotrienes, lipoxins and epoxyeicosanoids, exhibit a critical ability to regulate inflammation. It is therefore expected that the resolution of inflammation might be a valuable tool to prevent malignant transformation [135]. Moreover, it has been suggested that targeting the enzymes, such as PLA_2_, COX and LOX, would lead to the achievement of beneficial outcomes in cancer therapy [136].

The role of stromal sources of PG for pancreatic cancers has been demonstrated. Overexpression of COX-2 is a factor that links chronic inflammation with metaplastic and neoplastic changes in various tissues, including the exocrine pancreas. Indeed, elevated expression is associated with worse outcomes. The COX-2 product PGE2 acts through a receptor termed EP4, which has been found to be upregulated in cancer. Activation of this receptor supports cell proliferation, migration, invasion and metastasis through the activation of multiple signaling pathways, including extracellular signal-regulated kinase (ERK), cyclic adenosine monophosphate/protein kinase A (cAMP/PKA), phosphoinositide 3-kinase/protein kinase B (PI3K/AKT) and NFκB [137]. Stimulation of pancreatic cancer cells with PGE2 led to the secretion of fibroblast growth factor 1 (FGF1). This was related to the enhanced proliferation of cancer-associated fibroblasts and increased expression of vascular endothelial growth factor A (VEGF-A) [138]. In a transgenic murine model, cellular atypia and a loss of normal cell/tissue organization were observed. A diet containing celecoxib, a COX-2 inhibitor, prevented the development of an abnormal pancreatic phenotype [139]. Profibrogenic factors were upregulated in transgenic mice expressing COX-2 [76]. Another study reported that COX-1 expression was not detected in various pancreatic cancer cells and some of them also lacked COX-2 expression. This led the researchers to suspect that such cancers rely on exogenous sources of PG. Interestingly, fibroblasts expressing COX-1 and COX-2 might be a source of PG, which would be used by pancreatic cancer cells, allowing them to grow and to proliferate [114]. Pancreatic stellate cells (PSC) contribute, in addition to fibroblasts, as a source of fibrotic stroma [140]. Moreover, other components of the TME, such as the peripherical blood mononuclear cells, are able to produce PG [141]. A correlation between the levels of COX-2 and its product PGE2 and the extent of pancreatic fibrosis has been suggested. PGE2 stimulated the proliferation, migration and invasion of PSC. Furthermore, PGE2 increased the expression of extracellular matrix genes [142]. The peroxisome proliferator-activated receptor gamma (PPAR-γ) is a ligand-activated transcription factor that controls inflammation, in addition to growth and differentiation. In PSC, activation of this transcription factor inhibited cell proliferation and the synthesis of collagen [143]. Melatonin, the main product of the pineal gland, exhibits several pharmacological properties, including anti-inflammatory effects. In this context, this indolamine reduced the expression of COX-2 though the inhibition of NFκB signaling in pancreatic stellate cells subjected to hypoxia [144]. Melatonin has shown a significant antifibrotic role in the pancreas and some of its effects may be mediated by its anti-inflammatory actions exerted by modulating COX-2 expression. One of the characteristics of the pancreatic tumor microenvironment is the exclusion or poor infiltration of T cells [145]. This is one of the main reasons that pancreatic tumors are often refractory to immunotherapy. The production of PGE2 by the metabolization of AA via COX-2 appears to have an immunosuppressive effect, by which T cells are excluded from the tumor focus [146]. A similar observation was made by Zhang et al., who observed that the inhibition of COX-2 by apricoxib alone or in combination with anti-VEGF therapy increased the infiltration of CD8+ T cells in PDAC in vivo models [147].

Among the six different isoforms of LOX that have been described, 5-LOX is the most vital enzyme for the synthesis of leukotrienes. Evidence exists that relates 5-LOX to tumors [23]. Leukotriene signaling contributes to the active tumor microenvironment, promoting tumor growth and resistance to therapy [148]. Metabolites generated by 5-LOX from AA, such as 5-hydroxyeicosatetraenoic acid (5-HETE) and a variety of leukotrienes, have been suggested to act as mediators of inflammation in pathological states, leading to cancer. Furthermore, upregulation of the expression of 5-LOX has been associated with increased tumorigenesis. Pathways activated by 5-LOX may interact with the tumor microenvironment and can participate actively during the development and progression of a tumor [149]. Moreover, 5-LOX and the prostaglandin E synthase-1 (PGES-1) play an important role in the immune evasion evoked by tumor associated macrophages, which constrains the action of the antitumoral natural killer T cells [150].

Increased expression of 12-LOX and elevated levels of its metabolite 12-(S)-HETE were found in fibroblasts, one of the major nonmalignant cell types present in the stroma of pancreatic ductal adenocarcinoma (PDAC). This fact conferred upon fibroblasts the capability to transfer AA derivatives to tumor cells, which would be used for their proliferative needs [151]. However, not all metabolites of LOX exhibit fibrogenic effects. LXA4, a derivative metabolite of AA generated via LOX, inhibited the differentiation of PSC into a myofibroblast phenotype and reduced the proliferation and migration of pancreatic cancer cells evoked by PSC [152].

In addition to AA metabolism, CYP450 enzymes also are involved in the metabolism of drugs, foreign chemicals, cholesterol, steroids and other major lipids. This primarily takes place in the liver and in the gastrointestinal tract. However, evidence also exists that suggests that this occurs within the tumor microenvironment [153]. CYP450-derived AA epoxides, termed epoxyeicosatrienoic acids (EETs), also play a certain role in the promotion of the growth of certain tumors [154].

## 7. Discussion and Conclusions

As stated above, the relationship between inflammation and cancer is close. The existing literature covers a wide range of studies that show that isoenzymes of both COX and LOX are upregulated in different types of cancer, including pancreatic cancer. Less is known about the role of CYP450. Table 1 contains information about the functions of different metabolites of AA in pancreatitis, diabetes and pancreatic cancer The mentioned enzymes mainly convert AA into PGs, TXs, LTs and HETEs. These lipid-derived messengers are well-known mediators of intra- and intercellular signaling and are involved in cellular responses and fates. In this context, the role of the stroma and the putative cell-to-cell intercommunication in the pathophysiology of pancreatic diseases exhibits major relevance. The effect of certain inhibitors of the enzymes involved in AA metabolism is summarized in Table 2.

The stroma provides the conditions for the normal functioning of the cells present in a certain tissue or organ. Inflammation within the TME is a hallmark of cancer and is accepted as a major characteristic of carcinogens. Consequently, the stroma plays an important role in terms of a contribution to the development of malignant cells. In this context, the modulation of AA and the pathways regulated by its metabolites might be a valuable tool to prevent malignant transformation.

In conclusion, AA exhibits a role as an intracellular signal. Moreover, AA also acts as a precursor for eicosanoid synthesis, which further amplifies the actions of AA in terms of extending its actions to the surrounding cells. Cell communication within the TME depicts major relevance for cancer development and growth, and AA’s derivatives exhibit potential roles in the development of the disease. A summary of the main findings on the involvement of AA metabolization in the pancreatic tumor microenvironment is given in Figure 2.

## 8. Future Directions

In contrast to the wide variety of research focused on the involvement of COX and LOX in cancer development, fewer studies exist regarding the functions in carcinogenesis of CYP450. Therefore, further studies will be needed to shed more light on the role of CYP450 in pancreatic cancer development and/or progression. Additionally, although the majority of the studies reviewed point toward the deleterious actions of AA metabolism, protective actions of AA in the pancreas have been suggested, which need to be explored.

## Figures and Tables

**Figure 1 molecules-28-04316-f001:**
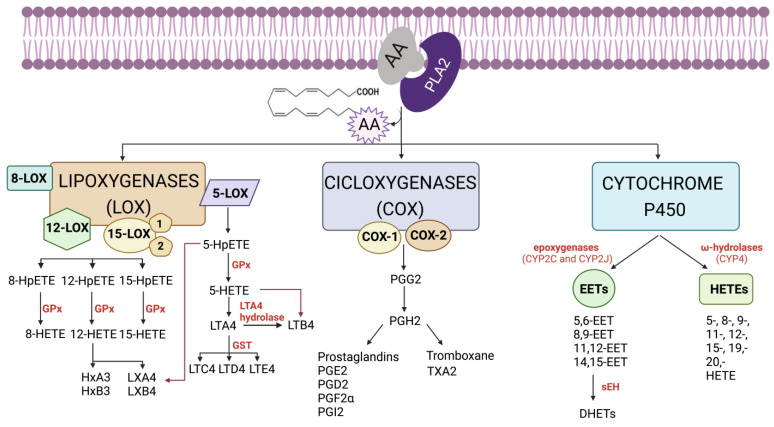
Schematic representation of the bioactive eicosanoids derived from arachidonic acid (AA) pathway. AA is mainly metabolized by three types of enzymes: cyclooxygenases (COX), lipoxygenases (LOX) and cytochrome P450 (CYP450). First, AA, which forms part of the phospholipids of the biological membrane, is released by phospholipase A2 (PLA_2_). Then, COX, LOX and/or CYP450 enzymes act on the free AA to generate a cascade of prostaglandins (PGs), thromboxanes (TXs), a series of hydroxyeicosatetraenoic acids (HETEs), leukotrienes (LTs), lipoxins (LXs) hepoxilins (HOs) and epoxyeicosatrienoics acids (EETs). Finally, the effect of each bioactive eicosanoid will depend on the specific receptor that it binds to (HpETE: hydroperoxy-eicosatetraenoic acid; GPx: glutathione peroxidase; GST: glutathione S-transferase; HxA3: 8-hydroxy-11,12-epoxy-eicosatetraenoic acid; HxB3: 10-hydroxy-11,12-epoxy-eicosatrienoic acid; sEH: soluble epoxidehydrolase; DHETs: dihydroxyepoxyeicosatrienoic acid). Created with BioRender.com (accessed on 17 May 2023).

**Figure 2 molecules-28-04316-f002:**
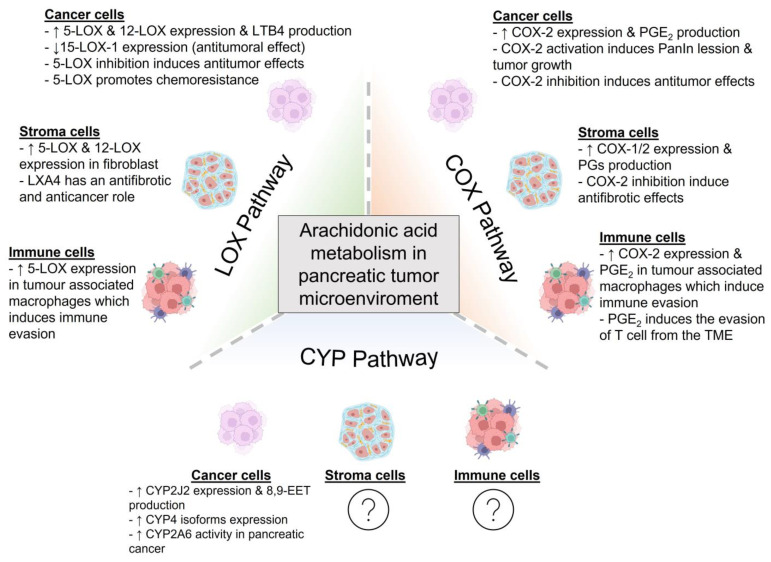
Metabolism of arachidonic acid in the pancreatic tumor microenvironment. The cells that form the pancreatic tumor microenvironment can metabolize arachidonic acid (AA) by the action of cyclooxygenases (COX), lipoxygenases (LOX) or cytochrome P450 (CYP450). These pathways give rise to different metabolites that can promote or diminish tumor development and growth. In this figure, the main findings on AA metabolization within the pancreatic tumor microenvironment are summarized (8,9-EET: 8,9-epoxyeicosatrienoic acid; COX: cyclooxygenase; LOX: lipoxygenase; CYP: cytochrome, LTB4: leukotriene B4; LXA4: lipoxin A4; PanIN: pancreatic intraepithelial neoplasia; PGE2: prostaglandin E2; PGs: prostaglandins; TME: tumor microenvironment). Created with BioRender.com (accessed on 17 May 2023).

**Table 1 molecules-28-04316-t001:** AA metabolites and its effects related to pancreatic diseases.

Diseases	AA Metabolism Pathway	Metabolite	Effect	Reference
**DIABETES**	**COX pathway**	PGE2	- β cell damage, dysfunction and final cell destruction in a murine model - Activates ERK, NFκB and JNK/STAT3 proinflammatory pathways in isolated rat pancreatic acini - Endogenous PGE2 acts as a protector against alcohol-induced injury in alcohol-fed rats	[3,79,104]
		TXB2	- Activates NFκB and JNK/STAT3 proinflammatory pathway in isolated rat pancreatic acini	[79]
	**LOX pathway**	12, 15-HETE	- Generates islet dysfunction and insulin resistance in prediabetic mice	[93]
	**CYP-450 pathway**	5,6-EET	- Increases insulin secretion from isolated rat pancreatic islets	[99]
		8,9-, 11,12- or 14,15-EET	- Increases glucagon secretion from isolated rat pancreatic islets	[99]
**PANCREATITIS**	**COX pathway**	TXB2, PGE2, 6-ketoPGF1α and PGD2	- High levels observed in chronic pancreatitis induced in rats, responsible for inflammation	[20,80]
		PGE2	- High levels observed in induced-acute pancreatitis, which produced edema, lipid peroxidation, inflammation, bleeding and pancreatic necrosis in animal models	[79,81,83]
		PGE2	- Promotes the activation of TNF-α in rat pancreatic acinar cells	[82]
	**LOX pathway**	LTB4 and 15-HETE	- Elevated levels have been observed in chronic pancreatitis in rats	[20]
		Lipoxins (LXs)	- Acts as anti-inflammatory mediator in the damage of tissue - Regulation of the expression of several transcription factors such as NFκB, EGR1, PPARγ, AP-1 in animal models - Interacts with many cells in both the innate and adaptive immune systems	[87]
		LTB4	- Generation of pancreatic edema, neutrophil infiltration and necrosis in rat pancreas - Produces polymorphonuclear leucocyte accumulation induced by oxygen free radicals in rats	[89,90]
**PANCREATIC CANCER**	**COX pathway**	PGE2	- Increases the cell proliferation of pancreatic tumor cells (MiaPaCa2) - Leads to FGF1 secretion by PANC-2 and this factor enhances proliferation of cancer-associated fibroblasts (CAFs) - Stimulates proliferation and migration of human PSC; also boosts the expression of extracellular matrix genes - Regulates the profibrotic activity of human PSC via EP4 receptor - Immunosuppressive effect that excludes T cells from pancreatic tumoral mass in human PDA	[114,138,142,146]
	**LOX pathway**	LXA4	- Inhibits the differentiation of human PSC into a CAF-like myofibroblast phenotype - Reduces the migration and growth of human pancreatic cell line - Decreases the growth of 3D heterospheroids of human PSC and PANC-1	[152]
		5,12-HETE	- Activation of the p44/42 and PI3/Akt pathway in cancer cells	[27]
	**CYP-450 pathway**	8,9-ETE	- Inhibit ferroptosis in the tumor line PANC-1	[34]
		20-HETE	- Overexpression in cancer promotes angiogenesis and metastasis via MMP activation in human pancreatic specimens	[126]

The table contains information about the functions of different metabolites of AA in pancreatitis, diabetes and pancreatic cancer. The metabolite is mentioned, in addition to the effects reported by researchers (reference include).

**Table 2 molecules-28-04316-t002:** Inhibitors of AA pathway and their effects on cell physiology.

Inhibitor	AA Metabolism Pathway	Therapeutic Target	Effector Function	Reference
**Rofecoxib**	**COX pathway**	COX-2	- Decreases PGE2 levels, collagen and TGFβ synthesis in a rat model	[84]
**Apricoxib**		COX-2	- Alone or combined with anti-VEGF, reduced the infiltration of CD8+ T cells in PDAC in vivo models	[147]
**Melatonin**		COX-2	- Decreases the expression of COX-2 via inhibition of NFκB pathway in rat PSC treated under hypoxia - Anti-inflammatory and antifibrotic actions in rat PSC	[144,145]
**5-aminosalicylic acid (compound of mesalazine)**		COX-2	- Inhibits PGE2 expression and NF-α- and IL-1β-induced COX-2 expression - Increases pancreatic duct permeability - Increases the risk of pancreatitis	[64]
**Nimesulide**		COX-2	- Reduced level of PGE2 and tumor angiogenesis in nude mice model (in vivo study)	[107]
**Ibuprofen**		COX	- Inhibits the development of pancreatic carcinogenesis induced in hamsters	[4]
**Celecoxib**		COX-2	- Prevents the development of abnormal pancreatic phenotype in mice	[139]
**MK-886**	**LOX pathway**	5-LOX activating protein (FLAP)	- Inhibitor of LT synthesis because it prevents the translocation of 5-LOX to the cell membrane in rats - Protective effects against pancreatic inflammation in cancer in animal models	[4,90,107]
**Zileuton**		5-LOX	- Reduces pancreatic damage and neutrophil activation in rat tissue	[91]
**Zafirlukast**		Leukotriene receptor antagonist	- Increases the pancreatitis histopathological parameters and necrosis in rats - Enhances insulin secretion in mice	[92,95]
**Zyflo**		5-LOX	- Increases the activity of antioxidant enzymes and reduces the concentration of lipid peroxidation in hamsters	[123,124]
**Triptolide**		5-LOX	- Cytotoxic effect on human pancreatic cell lines; induces apoptosis and reduces LTB4 production	[51]
**Protolichesterinic acid and lobaric acid**		5, 12-LOX	- Antiproliferative effects on pancreatic tumor cell lines Capan-1, Capan-2 and PANC-1	[122]

The table summarizes the effects of certain inhibitors of the enzymes involved in AA metabolism. The enzyme involved is mentioned, in addition to the effect induced on the cell type, as reported by scientists in their research (reference is cited).

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
