# Peer review of "Membrane Lipid Derivatives: Roles of Arachidonic Acid and Its Metabolites in Pancreatic Physiology and Pathophysiology"

_molecules, 2023, doi:10.3390/molecules28114316_

Round 1

Reviewer 1 Report

Thank you to the learned editor for choosing me as a reviewer. Although the article is within the scope of the journal, it needs major revisions

Correct the grammatical errors in the text.

Figure 1 and especially photo 2 lack proper quality.

In some cases, the indexes are not positioned correctly. be corrected For example, Ca+2

In review articles, it is necessary to present the studies in the form of a table to make them more understandable. Please provide one or two tables to summarize the content.

Mention the risk factors and causes of pancreatitis in general.

You can use the following sources to complete the content.

St13 protects against disordered acinar cell arachidonic acid pathway in chronic pancreatitis

Comparative Effect of the Active Substance of Thyme with N-Acetyl Cysteine on Hematological Parameters and Histopathological Changes of Bone Marrow and Liver in Rat Models of Acetaminophen Toxicity

The Antioxidant Properties of Alfalfa (Medicago sativa L.) and Its Biochemical, Antioxidant, Anti-Inflammatory, and Pathological Effects on Nicotine-Induced Oxidative Stress in the Rat Liver

The language of the article is good. But it needs some reforms.

Author Response

Thank you very much for reviewing our manuscript, for your suggestions and for drawing our attention to the different points raised.

Comment 1: Correct the grammatical errors in the text.

Reply: Thank you very much for this comment. We have proof read the text and corrected the grammar errors detected.

Comment 2: Figure 1 and especially photo 2 lack proper quality.

Reply: Thank you very much for this observation. The quality of the figures has been improved.

Comment 3: In some cases, the indexes are not positioned correctly. For example, Ca+2

Reply: Thank you very much for this observation. Indexes have been revised and positioned.

Comment 4: In review articles, it is necessary to present the studies in the form of a table to make them more understandable. Please provide one or two tables to summarize the content.

Reply: Following the reviewer´s suggestion we have included two tables summarizing the content, i.e., the content about the manuscripts used in the review.

Comment 5: Mention the risk factors and causes of pancreatitis in general.

Reply: Thank you very much for this suggestion. The risk factors and causes of pancreatitis have been mentioned. New text and corresponding references have been given in the paragraph corresponding to “Role of arachidonic acid in pancreatitis and diabetes”.

Comment 6: You can use the following sources to complete the content.

St13 protects against disordered acinar cell arachidonic acid pathway in chronic pancreatitis

Comparative Effect of the Active Substance of Thyme with N-Acetyl Cysteine on Hematological Parameters and Histopathological Changes of Bone Marrow and Liver in Rat Models of Acetaminophen Toxicity

The Antioxidant Properties of Alfalfa (Medicago sativa L.) and Its Biochemical, Antioxidant, Anti-Inflammatory, and Pathological Effects on Nicotine-Induced Oxidative Stress in the Rat Liver

Reply: Thank you very much for this comment. We have included in the text the reference on St3 role in inflammation, suggested by the reviewer, which is related with pancreatic damage and AA involvement in pancreatic inflammation. Since the manuscript is focused onto pancreatic inflammation, we considered that studies on liver inflammation and/or toxicity are not directly related to the topic and, therefore, we did not include some of the above suggested references.

Comment 7: Comments on the Quality of English Language. The language of the article is good. But it needs some reforms.

Reply: Thank you very much for this observation. English has been revised.

Reviewer 2 Report

In this review, Ortiz-Placín et al. presented the physiopathological roles of arachidonic acid and its metabolites in the pancreas, with a special focus on pancreatic diseases including pancreatitis, diabetes, and pancreatic tumor and its surrounding stroma. The review is well-organized and well-written. The reviewer suggests accepting it in its current format if the authors could increase the font size in both figures. 

Author Response

Thank you very much for reviewing our manuscript, for the time given to our work and for your suggestion.

Comment: In this review, Ortiz-Placín et al. presented the physiopathological roles of arachidonic acid and its metabolites in the pancreas, with a special focus on pancreatic diseases including pancreatitis, diabetes, and pancreatic tumor and its surrounding stroma. The review is well-organized and well-written. The reviewer suggests accepting it in its current format if the authors could increase the font size in both figures.

Reply:

We have increased the font size in both figures.

Reviewer 3 Report

Manuscriot entitled "Membrane lipid´s derivatives: roles of arachidonic acid and its metabolites in pancreatic diseases."

This work is well drafted and the figures are of high quality. This work should be accepted for publication pending minor revision as follows:

1. The clinical relevance of arachidonic acid and its metabolites, as well as associated genes shouldbe added. For example, the biology, prognostication, therapeutic aspects of these issues in pancreatic diseases should be mentioned.

2. The word "pancreatic diseases" is not focusing. The authors should focus on certain area, that could be cancer or inflammaiton.

acceptable

Author Response

Thank you very much for reviewing our manuscript, for your suggestions and for drawing our attention to the different points raised.

Comment 1: The clinical relevance of arachidonic acid and its metabolites, as well as associated genes should be added. For example, the biology, prognostication, therapeutic aspects of these issues in pancreatic diseases should be mentioned.

Reply: Thank you very much for this comment. Following the reviewer´s suggestion we have included in the manuscript information regarding the biology, prognostication, therapeutic aspects of AA and its metabolites in pancreatic diseases. The text, and the corresponding references, have been included at the end of the paragraph dedicated to “Arachidonic acid metabolism in the pancreas”.

Comment 2: The word "pancreatic diseases" is not focusing. The authors should focus on certain area, that could be cancer or inflammation.

Reply: Thank you very much for this observation. We have changed the term “pancreatic diseases” and instead we have now written “pancreatic physiology and pathophysiology” in the title of the manuscript. In the rest of the manuscript, the term “pancreatic diseases” has been changed. Because the manuscript was initially designed as a review about the involvement of arachidonic acid and its metabolites in pancreatic physiology and pathophysiology, if used, the mentioned term refers generally to pancreatic cancer, pancreatitis and/or diabetes on the whole, as pancreatic disorders. Then there are chapters referred to each disorder specifically.

Round 2

Reviewer 1 Report

correction have been made.